# Male-predominant galanin mediates androgen-dependent aggressive chases in medaka

**Junpei Yamashita[1], Akio Takeuchi[1], Kohei Hosono[1†], Thomas Fleming[1], Yoshitaka Nagahama[2], Kataaki Okubo[1]***

[1]Department of Aquatic Bioscience, Graduate School of Agricultural and Life Sciences, The University of Tokyo, Tokyo, Japan; [2]Division of Reproductive Biology, National Institute for Basic Biology, Okazaki, Japan

**Abstract** Recent studies in mice demonstrate that a subset of neurons in the medial preoptic area (MPOA) that express galanin play crucial roles in regulating parental behavior in both sexes. However, little information is available on the function of galanin in social behaviors in other species. Here, we report that, in medaka, a subset of MPOA galanin neurons occurred nearly exclusively in males, resulting from testicular androgen stimulation. Galanin-deficient medaka showed a greatly reduced incidence of male–male aggressive chases. Furthermore, while treatment of female medaka with androgen induced male-typical aggressive acts, galanin deficiency in these females attenuated the effect of androgen on chases. Given their male-biased and androgen-dependent nature, the subset of MPOA galanin neurons most likely mediate androgen-dependent male–male chases. Histological studies further suggested that variability in the projection targets of the MPOA galanin neurons may account for the species-dependent functional differences in these evolutionarily conserved neural substrates.

*For correspondence:
okubo@marine.fs.a.u-tokyo.ac.jp

Present address: [†]School of Life Science and Technology, Tokyo Institute of Technology, Yokohama, Japan

Competing interests: The authors declare that no competing interests exist.

## Introduction

Almost all animals interact socially with conspecifics at some stage of their lives (e.g. for territorial/resource disputes, mating, and parenting) (*Hofmann et al., 2014*; *Chen and Hong, 2018*). A group of reciprocally connected brain regions that regulates social behaviors in concert with sex steroids and neuropeptides, sometimes referred to as the 'social behavior network', has been highly conserved over the 450 million years of vertebrate evolution (*Newman, 1999*; *Goodson, 2005*; *O'Connell and Hofmann, 2012*). The preoptic area (POA) is the most highly conserved node of this network, containing subsets of steroid receptor- and neuropeptide-expressing neurons that regulate a wide range of social behaviors (*O'Connell and Hofmann, 2012*). While the social behavior network seems to be fundamental to the expression of social behaviors in all vertebrate species and both sexes, a large variation in social behaviors is apparent across species and sexes. This suggests that there is a degree of underlying variation—at either the structural or chemical level—in behaviorally relevant neural circuits, which in turn has spurred a growing interest in uncovering the neural substrates underlying species- and sex-dependent differences in social behaviors (*Yang and Shah, 2014*; *Hofmann et al., 2014*; *Chen and Hong, 2018*).

Recent evidence in mice demonstrates that a subset of neurons in the medial POA (MPOA) that express the pleiotropic neuropeptide galanin (abbreviated as GAL or Gal depending on species) play a crucial role in regulating parental behavior in both sexes (*Wu et al., 2014*; *Kohl et al., 2018*). GAL-expressing neurons are commonly found in the MPOA of vertebrates, including teleost fish, the majority of which do not provide parental care (*Fischer and O'Connell, 2017*). This raises the question of the role of these neurons in such non-parental species. Curiously, studies in several teleost

species have consistently shown that males have many more MPOA Gal neurons than females (*Cornbrooks and Parsons, 1991*; *Rao et al., 1996*; *Rodríguez et al., 2003*; *Tripp and Bass, 2020*). This contrasts with the situation in mammals, where either there is no sex difference in the number of these neurons, or the direction of the sex difference (if any) varies among species and even within strains in mice (*Bloch et al., 1993*; *Park et al., 1997*; *Mathieson et al., 2000*; *Wu et al., 2014*). Based on this distinctive feature, Gal neurons in the teleost MPOA are assumed to be involved in some male-biased behavior. Indeed, studies in midshipman fish (*Porichthys notatus*) have shown that MPOA Gal neurons are activated during mating, thereby suggesting that they have a role in mating behavior (*Tripp and Bass, 2020*; *Tripp et al., 2020*). To our knowledge, however, no attempt has been made to directly test these ideas. Furthermore, the mechanism that causes the male-biased sex difference in teleost Gal neurons remains to be elucidated.

In recent years, the teleost fish medaka (*Oryzias latipes*) has emerged as an effective model organism for studying social behaviors (*Okuyama et al., 2014*; *Hiraki-Kajiyama et al., 2019*; *Yokoi et al., 2020*). In medaka, females spawn once daily over the spawning period, whereas males can spawn multiple times a day (*Weir and Grant, 2010*). Neither males nor females provide parental care after spawning. Adult males compete aggressively for territories and access to food and mates, whereas females engage in far less aggression (*Grant and Foam, 2002*). Here, we have identified a specific population of neurons in the medaka MPOA that express *gal* nearly exclusively in males, extending the findings in other teleost species. We have further explored the mechanism underlying this differential *gal* expression and provide direct evidence that Gal indeed has a male-biased, but unexpected, behavioral role in medaka.

## Results

### Male-biased sexual dimorphism exists in *gal* expression in the MPOA

Screening of a full-length cDNA library constructed from the medaka brain facilitated the isolation of a cDNA whose deduced protein had the best BLAST hits to GAL in other species (deposited in GenBank with accession number LC532140) (*Figure 1—figure supplement 1*). Phylogenetic tree analysis confirmed that this protein represented the medaka ortholog of GAL (*Figure 1A*).

Northern blot analysis revealed the presence of a 0.8 kb *gal* transcript, which was much more abundant in the adult male brain than in the female brain (*Figure 1B*). This result was confirmed by quantitative real-time PCR analysis, which showed a male-biased sexual dimorphism in *gal* expression in the adult brain ($p<0.0001$, $t = 19.3$, $df = 7.12$; unpaired *t*-test with Welch's correction) (*Figure 1C*).

We used in situ hybridization to define the brain nucleus responsible for the sexual dimorphism in *gal* expression. Neurons expressing *gal* were observed in the anterior part of the parvocellular portion of the magnocellular preoptic nucleus (aPMp)/anterior parvocellular preoptic nucleus (PPa), the posterior part of the parvocellular portion of the magnocellular preoptic nucleus (pPMp), and the posterior parvocellular preoptic nucleus (PPp) in the MPOA, and the anterior tuberal nucleus (NAT)/ventral tuberal nucleus (NVT)/lateral recess nucleus (NRL) in the hypothalamus of the adult brain of both sexes (*Figure 1D–G*; abbreviations for medaka brain nuclei are given in *Supplementary file 1*). Faint-to-moderate expression of *gal* was also seen in the pineal gland (data not shown). Although there were no sex differences in these spatial patterns, striking sexual dimorphism was observed in pPMp, where *gal*-expressing neurons occurred much more abundantly in males than in females ($p<0.0001$, $t = 12.1$, $df = 8$; unpaired *t*-test) (*Figure 1D and G*). In addition to pPMp, NAT/NVT/NRL contained a slightly higher, but comparable, number of *gal*-expressing neurons in males than in females ($p=0.047$, $t = 2.35$, $df = 8$; unpaired *t*-test) (*Figure 1D*). Collectively, these results indicate that pPMp in the MPOA is the major source of the sexual dimorphism in overall *gal* expression in the medaka brain.

Next, we explored when the sexual dimorphism in brain *gal* expression is established during growth and sexual maturation. Examination of brains at different ages by real-time PCR revealed that the male bias in *gal* expression increased steadily with age (main effect of sex: $F_{(1, 56)}=464$, $p<0.0001$; main effect of age: $F_{(3, 56)}=180$, $p<0.0001$; and interaction between sex and age: $F_{(3, 56)}=106$, $p<0.0001$) (*Figure 1H*). *gal* was expressed at significantly higher levels in male than in female brains as early as 2 months of age ($p=0.0002$, $t = 4.39$, $df = 56$; Bonferroni's post hoc test),

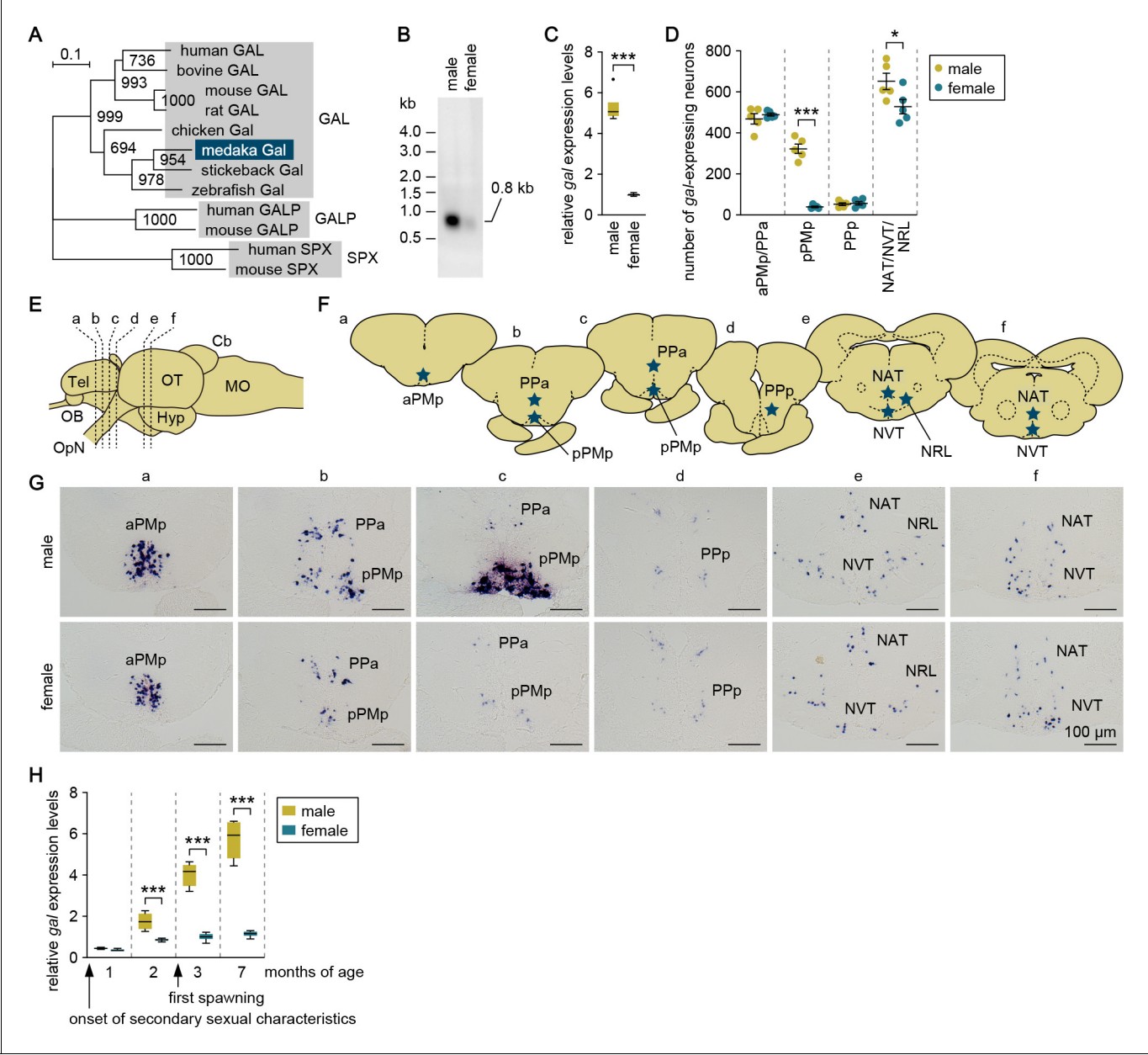

**Figure 1.** Male-biased sexual dimorphism exists in *gal* expression in the MPOA. (**A**) Phylogenetic tree showing the relationship of medaka Gal to other known GAL family proteins. The number at each node indicates bootstrap values for 1000 replicates. Scale bar represents 0.1 substitutions per site. GALP, galanin-like peptide; SPX, spexin hormone. For species names and GenBank accession numbers, see ***Supplementary file 2***. (**B**) Detection of *gal* transcript in the whole brain of adult males and females by Northern blot analysis. Sizes (in kb) of RNA markers and the detected band are indicated on the left and right, respectively. (**C**) Confirmation of male-biased expression of *gal* in adult whole brain by real-time PCR (n = 8 per sex). ***p<0.001 (unpaired *t*-test with Welch's correction). (**D**) Number of *gal*-expressing neurons in each brain nucleus of adult males and females (n = 5 per sex). *p<0.05; ***p<0.001 (unpaired *t*-test). (**E**) Line drawing of a lateral view (anterior to the left) of the medaka brain showing the approximate levels of sections in panel G. (**F**) Line drawing of coronal sections showing the location of brain nuclei containing *gal*-expressing neurons (stars). (**G**) Representative micrographs showing *gal*-expressing neurons in each brain nucleus of adult males (upper panels) and females (lower panels). Scale bars represent 100 μm. (**H**) Levels of *gal* expression in the whole brain of males and females during growth and sexual maturation (n = 8 per sex and stage). ***p<0.001 (Bonferroni's post hoc test). For abbreviations of brain regions and nuclei, see ***Supplementary file 1***. See also ***Figure 1—figure supplement 1***.

The online version of this article includes the following figure supplement(s) for figure 1:

**Figure supplement 1.** Sequence information for medaka *gal*.

when secondary sexual characteristics were well-developed but spawning had not yet occurred (spawning started at 3 months). No significant sex difference in brain *gal* expression was observed at an earlier age (1 month), when secondary sexual characteristics began to appear.

## Sexually dimorphic *gal* expression is dependent on adult sex steroids

We explored the mechanisms underlying the sexual dimorphism in *gal* expression. First, we estimated the magnitude of chromosomal and gonadal influences on the pattern of *gal* expression by producing sex-reversed medaka and examining *gal* expression in the brain. Real-time PCR analysis revealed that sex-reversed XX males exhibited the same high level of brain *gal* expression as typical XY males, whereas sex-reversed XY females showed a much lower level ($p<0.0001$, $t = 17.2$, $df = 28$ versus XY males; $p<0.0001$, $t = 15.8$, $df = 28$ versus XX males; Bonferroni's post hoc test), comparable to that in typical XX females (*Figure 2A*). Consistent with these results, in situ hybridization analysis showed that the number of *gal*-expressing neurons in each brain nucleus of sex-reversed XX males and XY females was roughly equivalent to that in typical XY males and XX females, respectively. XX males had many more *gal*-expressing neurons in pPMp ($p<0.0001$, $t = 8.71$, $df = 8$; unpaired *t*-test) and slightly more *gal*-expressing neurons in NAT/NVT/NRL ($p=0.048$, $t = 2.33$, $df = 8$; unpaired *t*-test) as compared with XY females (*Figure 2B and C*). These results indicate that the pattern of brain *gal* expression is independent of sex chromosome complement but linked to gonadal phenotype.

These findings led us to hypothesize that the gonadal sex steroid milieu at puberty and during adulthood has a major impact on the pattern of brain *gal* expression. To test this idea, we analyzed brain *gal* expression in fish that were gonadectomized as adults and treated with 11-ketotestosterone (KT; the primary, non-aromatizable androgen in teleosts) or estradiol-17β (E2; the major estrogen in vertebrates including teleosts). In males, the level of brain *gal* expression, as measured by real-time PCR, was significantly reduced by castration ($p=0.016$, $t = 3.04$, $df = 27$; Bonferroni's post hoc test), and restored by treatment with KT ($p<0.0001$, $t = 6.74$, $df = 27$; Bonferroni's post hoc test) but not E2 ($p>0.99$, $t = 0.679$, $df = 27$; Bonferroni's post hoc test) (*Figure 2D*). In accordance with these results, in situ hybridization revealed that castration caused a significant reduction in the number of *gal*-expressing neurons in pPMp ($p=0.0048$, $t = 3.79$, $df = 16$; Bonferroni's post hoc test), which was restored by KT treatment ($p=0.0052$, $t = 3.75$, $df = 16$; Bonferroni's post hoc test) but not by E2 treatment ($p=0.28$, $t = 1.79$, $df = 16$; Bonferroni's post hoc test) (*Figure 2E and F*), although E2 treatment increased the number of *gal*-expressing neurons in aPMp/pPPa ($p=0.013$, $t = 3.33$, $df = 16$; Bonferroni's post hoc test) (*Figure 2E*). In females, no significant changes in the level of brain *gal* expression after ovariectomy or steroid treatments were detected by real-time PCR (*Figure 2G*). However, in situ hybridization revealed that ovariectomy caused a significant increase in the number of *gal*-expressing neurons in pPMp ($p=0.020$, $t = 3.11$, $df = 16$; Bonferroni's post hoc test), which was further enhanced by subsequent KT treatment ($p=0.0008$, $t = 4.63$, $df = 16$; Bonferroni's post hoc test) (*Figure 2H and I*). Conversely, the increase was significantly attenuated by E2 treatment ($p=0.040$, $t = 2.79$, $df = 16$; Bonferroni's post hoc test) (*Figure 2H and I*). Together, these results indicate that high circulating levels of androgen derived from the testis cause a marked increase in *gal*-expressing neurons in pPMp in adult males, whereas the absence of androgen stimulation in adult females means that these neurons remain at basal levels. In addition, these neurons are diminished in adult males deprived of androgen and, conversely, are induced in adult females deprived of estrogen and supplemented with androgen.

To assess whether androgen and estrogen could exert direct actions on sexually dimorphic *gal*-expressing neurons in pPMp, we analyzed gene expression of the androgen receptor (AR; *ara* and *arb*) and estrogen receptor (ER; *esr1*, *esr2a*, and *esr2b*) in these neurons by double in situ hybridization. *ara* was expressed in a large proportion of *gal*-expressing neurons in males and in some of the sparse *gal*-expressing neurons in females, whereas none of these neurons were positive for *arb* expression in either sex (*Figure 2—figure supplement 1*). *esr1* was expressed in a large proportion of *gal*-expressing neurons in both sexes, whereas *esr2a* and *esr2b* were expressed in these neurons only in males (*Figure 2—figure supplement 1*). These results suggest that androgen and estrogen may directly act on *gal*-expressing neurons in pPMp and that, if so, the stimulatory action of androgen is mediated via *ara*, while the inhibitory action of estrogen seen in females is mediated via *esr1*.

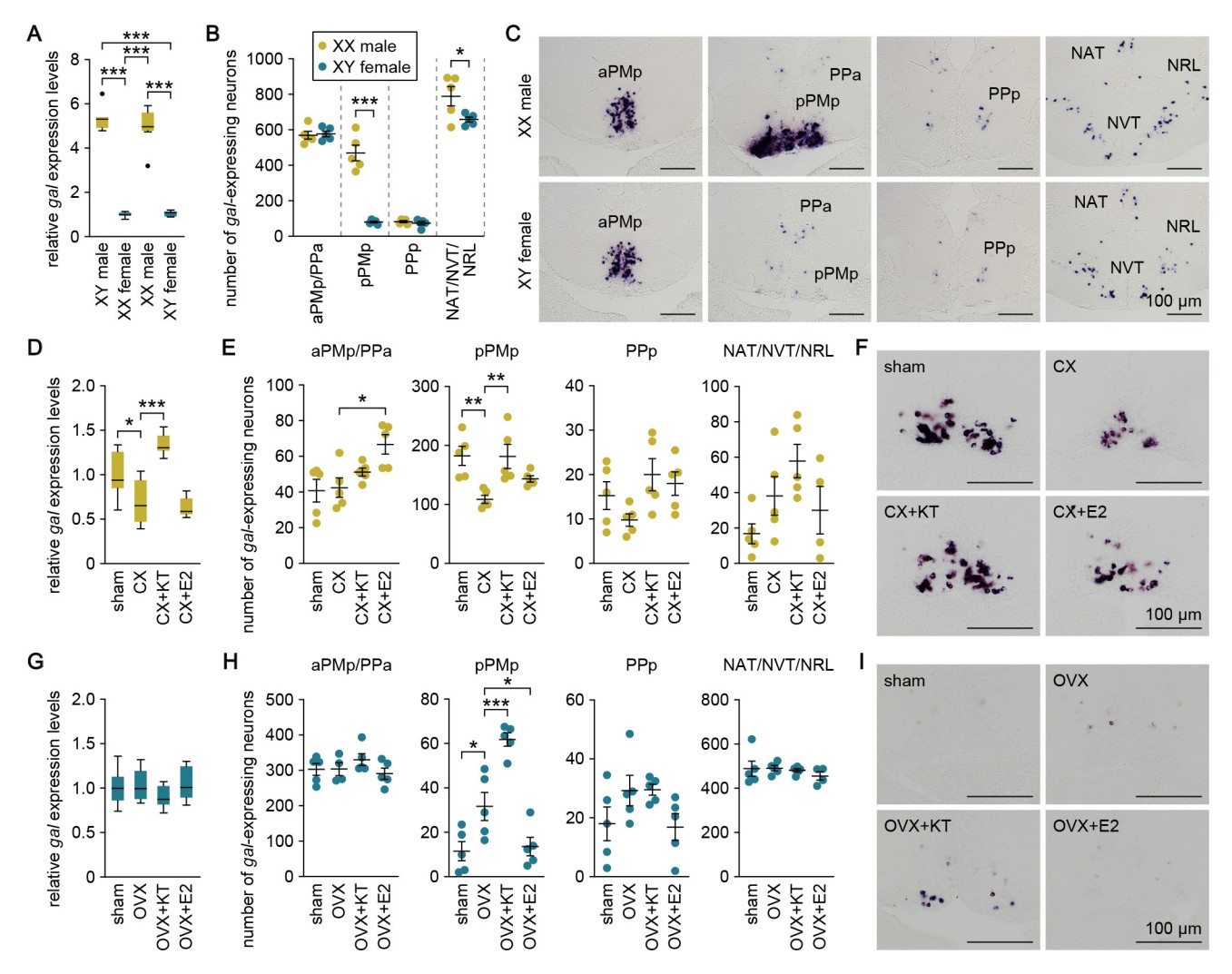

**Figure 2.** Sexually dimorphic *gal* expression is dependent on adult sex steroids. (A) Levels of *gal* expression in the whole brain of sex-reversed adult XX males and XY females versus typical adult XY males and XX females (n = 8 per group). ***p<0.001 (Bonferroni's post hoc test). (B) Number of *gal*-expressing neurons in each brain nucleus of sex-reversed adult XX male and XY females (n = 5 per group). *p<0.05; ***p<0.001 (unpaired *t*-test). (C) Representative micrographs showing *gal*-expressing neurons in each brain nucleus of sex-reversed adult XX males (upper panels) and XY females (lower panels). (D) Levels of *gal* expression in the whole brain of sham-operated males (sham; n = 8) and castrated males exposed to vehicle alone (CX; n = 8), KT (CX+KT; n = 9), or E2 (CX+E2; n = 6). *p<0.05; ***p<0.001 (Bonferroni's post hoc test). (E) Number of *gal*-expressing neurons in each brain nucleus of sham, CX, CX+KT, and CX+E2 males (n = 5 per group). *p<0.05; **p<0.01 (Bonferroni's post hoc test). (F) Representative micrographs showing *gal* expression in pPMp of sham, CX, CX+KT, and CX+E2 males. (G) Levels of *gal* expression in the whole brain of sham females and ovariectomized females exposed to vehicle alone (OVX), KT (OVX+KT), or E2 (OVX+E2) (n = 12 per group). (H) Number of *gal*-expressing neurons in each brain nucleus of sham, OVX, OVX+KT, and OVX+E2 females (n = 5 per group). *p<0.05; ***p<0.001 (Bonferroni's post hoc test). (I) Representative micrographs showing *gal* expression in pPMp of sham, OVX, OVX+KT, and OVX+E2 females. Scale bars represent 100 μm. For abbreviations of brain nuclei, see *Supplementary file 1*. See also *Figure 2—figure supplement 1*.

The online version of this article includes the following figure supplement(s) for figure 2:

**Figure supplement 1.** Expression of sex steroid receptors in sexually dimorphic *gal* neurons in pPMp.

## Genetic ablation of *gal* specifically suppresses male–male chases

The above results, together with the behavioral effects of Gal observed in mice, led us to assume that the male-predominant expression of *gal* may mediate a male-typical behavior in medaka. To investigate this possibility, we generated *gal* knockout medaka by using clustered regularly inter-spaced short palindromic repeats (CRISPR)/CRISPR-associated protein 9 (Cas9)-mediated genome

editing (*Figure 3—figure supplement 1*) and examined mating behavior and intrasexual aggressive behavior in both sexes. Two independently derived knockout lines (Δ2 and Δ10) were tested to ensure the reproducibility of the results and to eliminate the possibility of off-target effects of CRISPR/Cas9. Successful ablation of *gal* was confirmed by the complete loss of Gal immunoreactivity in homozygous knockout brains from both lines (*Figure 3—figure supplement 1*); the specificity of the anti-GAL antibody used was verified by simultaneous application of immunohistochemistry and in situ hybridization, which showed that the antibody selectively labeled neuronal cell bodies expressing the *gal* transcript (*Figure 3—figure supplement 1*).

The mating behavior tests revealed that most heterozygous and homozygous knockout males from the Δ2 and Δ10 lines spawned successfully, similar to wild-type males. Moreover, there were no significant differences in any of the detailed parameters of mating behavior among wild-type, heterozygous knockout, and homozygous knockout males of either line (*Figure 3—figure supplement 2*). Similarly, females did not show significant differences in any of the parameters of mating behavior among genotypes (*Figure 3—figure supplement 2*).

The aggressive behavior tests revealed that heterozygous and homozygous knockout males from the Δ2 and Δ10 lines exhibited significantly fewer chases as compared with wild-type males (p=0.0003, $t$ = 5.31, $df$ = 14 for +/Δ2; p=0.0020, $t$ = 4.36, $df$ = 14 for Δ2/Δ2; p=0.0087, $t$ = 3.55, $df$ = 15 for +/Δ10; and p=0.012, $t$ = 3.40, $df$ = 15 for Δ10/Δ10; Bonferroni's post hoc test) (*Figure 3A and B*). Examination of the number of chases in each 5 min interval during the 20-min test period revealed that homozygous knockout males performed fewer chases as compared with wild-type males in all time intervals, with significant differences in the first 10 min for the Δ2 line (main effect of genotype: F (1, 10)=27.2, p=0.0004; main effect of time: F (3, 30)=1.05, p=0.38; interaction between genotype and time: F (3, 30)=2.29, p=0.098; p=0.0004, $t$ = 4.31, $df$ = 40 at 0–5 min; p=0.017, $t$ = 3.03, $df$ = 40 at 5–10 min; p=0.73, $t$ = 1.36, $df$ = 40 at 10–15 min; and p>0.99, $t$ = 0.957, $df$ = 40 at 15–20 min; Bonferroni's post hoc test) (*Figure 3C*) and in the last 10 min for the Δ10 line (main effect of genotype: F (1, 10)=12.3, p=0.0057; main effect of time: F (3, 30)=8.62, p=0.0003; interaction between genotype and time: F (3, 30)=3.07, p=0.043; p>0.99, $t$ = 0.519, $df$ = 40 at 0–5 min; p=0.28, $t$ = 1.87, $df$ = 40 at 5–10 min; p=0.0013, $t$ = 3.94, $df$ = 40 at 10–15 min; and p=0.014, $t$ = 3.11, $df$ = 40 at 15–20 min; Bonferroni's post hoc test) (*Figure 3D*). Further analysis of the number of chases initiated and received by each individual male revealed that not a few but rather many males both initiated and received chases regardless of their genotype (*Figure 3E and F*).

The +/Δ2 and Δ2/Δ2 males also showed significantly fewer fin displays relative to wild-type males (p=0.0028, $t$ = 4.18, $df$ = 14 for +/Δ2; p=0.0067, $t$ = 3.73, $df$ = 14 for Δ2/Δ2; Bonferroni's post hoc test); however, no similar trend was observed for Δ10 males, thus, there was a lack of reproducibility between the lines (*Figure 3A and B*). There were no significant differences in the number of circles, strikes, or bites among males of any genotype (*Figure 3A and B*). In contrast to males, females of all genotypes including wild-type exhibited no or very few aggressive acts, and there were no significant differences in any of the acts evaluated among them (*Figure 3G and H*).

In summary, genetic ablation of *gal* in medaka resulted in a specific behavioral deficit in male–male chases, indicating that *gal* plays a role exclusively in this aggressive behavioral act.

## Genetic ablation of *gal* attenuates androgen-induced chases in females

Our preliminary data indicated that female medaka display male-typical aggressive behaviors, including chases, when treated with androgen as adults. To verify that *gal* specifically affects chases and that this process is mediated by the male-biased, androgen-dependent population of *gal*-expressing neurons in pPMp, we treated adult females from the *gal* knockout lines with KT and evaluated their aggressive behaviors. Androgen treatment significantly increased the number of chases, fin displays, and bites in the two lines (main effect of androgen treatment: F (1, 20)=33.7, p<0.0001 and F (1, 20)=177, p<0.0001 for chases in Δ2 and Δ10, respectively; F (1, 20)=3.62, p=0.072 and F (1, 20)=14.4, p=0.0011 for fin displays in Δ2 and Δ10, respectively; and F (1, 20)=26.6, p<0.0001 and F (1, 20)=313, p<0.0001 for bites in Δ2 and Δ10, respectively) (*Figure 4A and B*). Although there were no significant between-genotype differences in fin displays or bites, the homozygous knockout females performed significantly fewer chases relative to wild-type females (main effect of genotype: F (1, 20)=10.2, p=0.0046 and F (1, 20)=36.9, p<0.0001 in Δ2 and Δ10, respectively; interaction between androgen treatment and genotype: F (1, 20)=11.5, p=0.0029 and F (1, 20)=36.9, p<0.0001

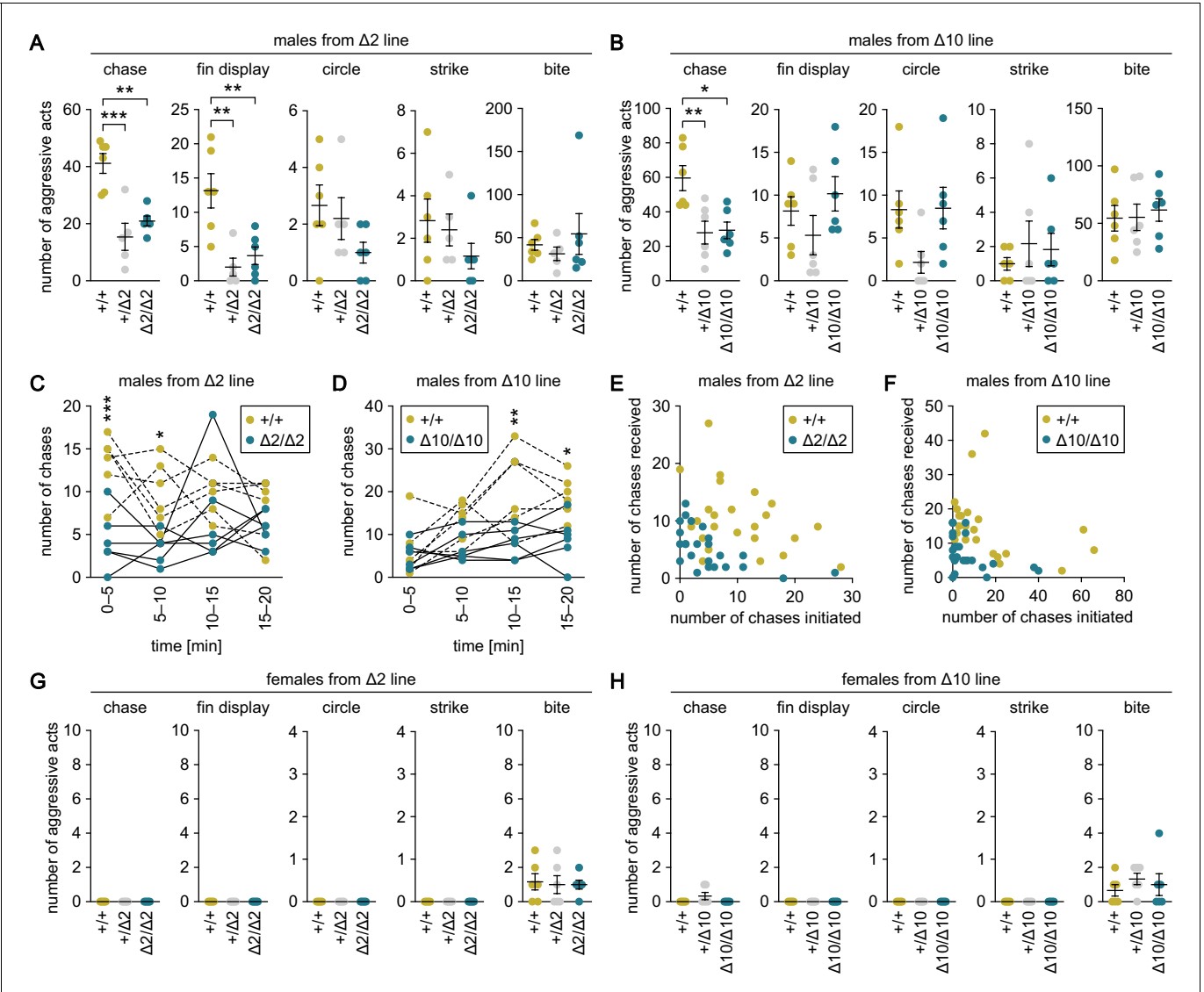

**Figure 3.** Genetic ablation of *gal* specifically suppresses male–male chases. (A, B) Sum of each aggressive behavioral act (chase, fin display, circle, strike, and bite) performed by wild-type (+/+), heterozygous (+/Δ2 and +/Δ10), and homozygous (Δ2/Δ2 and Δ10/Δ10) males from Δ2 (A) and Δ10 (B) *gal* knockout lines. n = 6 per group, except +/Δ2 males, where n = 5. **p<0.01; ***p<0.001 (Bonferroni's post hoc test). (C, D) Number of chases performed by wild-type and homozygous males from Δ2 (C) and Δ10 (D) *gal* knockout lines for each 5-min interval. n = 6 per genotype. Asterisks indicate significant differences between the two genotypes in the same time interval. *p<0.05; **p<0.01; ***p<0.001 (Bonferroni's post hoc test). (E, F) Number of chases initiated (x-axis) and received (y-axis) by each wild-type and homozygous male from Δ2 (E) and Δ10 (F) *gal* knockout lines. (G, H) Sum of each aggressive behavioral act (chase, fin display, circle, strike, and bite) performed by wild-type, heterozygous, and homozygous females from Δ2 (G) and Δ10 (H) *gal* knockout lines. n = 6 per group. See also *Figure 3—figure supplement 1* and *Figure 3—figure supplement 2*.

The online version of this article includes the following figure supplement(s) for figure 3:

**Figure supplement 1.** Generation and verification of *gal* knockout medaka.

**Figure supplement 2.** Mating behavior of *gal* knockout medaka.

in Δ2 and Δ10, respectively; and Bonferroni's post hoc test: p=0.0003, *t* = 4.65, *df* = 20 and p<0.0001, *t* = 8.60, *df* = 20 in Δ2 and Δ10, respectively) (*Figure 4A and B*).

Immediately after the behavioral tests, the brains were analyzed for the presence of Gal-expressing neurons in pPMp by immunohistochemistry. As expected, Gal-immunoreactive neurons were apparent in pPMp of wild-type females, but not homozygous knockout females (*Figure 4C*). These results demonstrate that androgen induces male-typical aggressive behaviors in females, but genetic ablation of *gal* specifically attenuates the effect of androgen on chases.

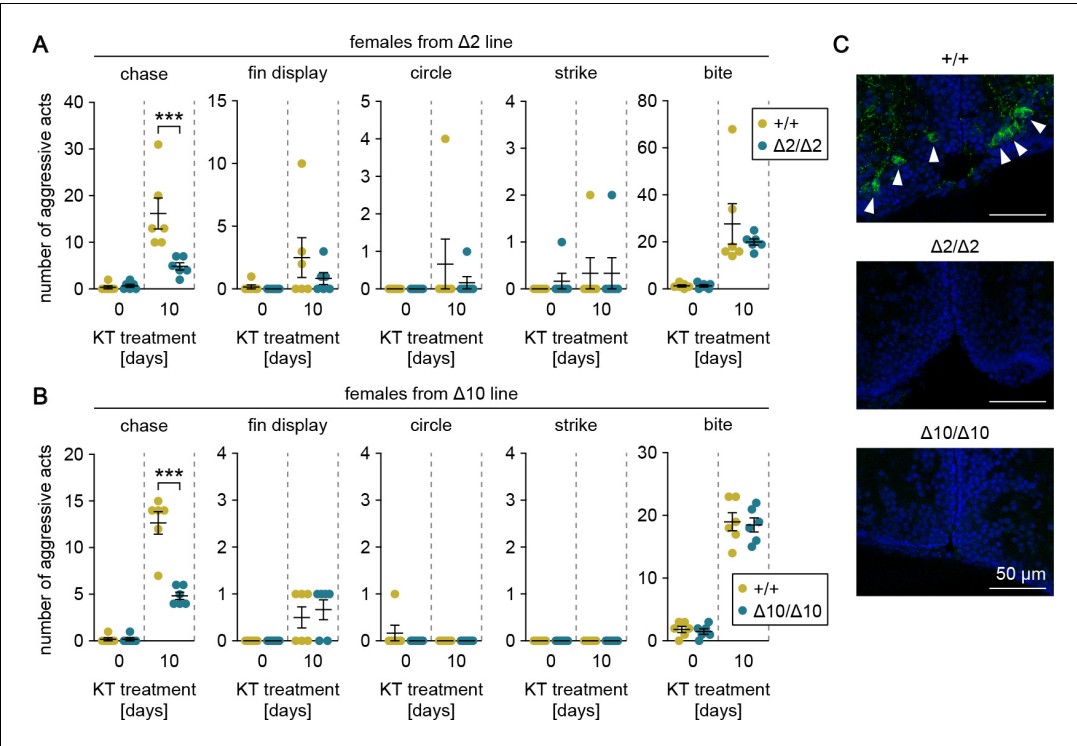

**Figure 4.** Genetic ablation of *gal* attenuates androgen-induced chases in females. (**A, B**) Sum of each aggressive behavioral act (chase, fin display, circle, strike, and bite) performed by KT-treated females from Δ2 (**A**) and Δ10 (**B**) *gal* knockout lines. Asterisks indicate significant differences between wild-type (+/+) and homozygous (Δ2/Δ2 and Δ10/Δ10) genotypes on the same day. n = 6 per group. ***p<0.001 (Bonferroni's post hoc test). (**C**) Representative micrographs of coronal pPMp sections from KT-treated females of Δ2 and Δ10 *gal* knockout lines showing the induction of Gal-expressing neurons in wild-type but not homozygous knockout females. Arrowheads indicate Gal-immunoreactive (green) neuronal cell bodies. Blue color indicates nuclear counterstaining. Scale bars represent 50 μm.

## Gal peptide produced male-predominantly is transported to various brain regions

Next, we investigated the sites of action of the Gal polypeptide that is produced male-predominantly in pPMp. First, we determined the axonal projections of *gal*-expressing neurons, and compared them between the sexes by immunohistochemistry. Dense plexuses of Gal-immunoreactive axons were observed in the ventrorostral part of the telencephalon, MPOA, hypothalamus, and pituitary of both sexes (*Figure 5*), suggesting that these axons arise from *gal*-expressing neurons in aPMp/PPa, PPp, and NAT/NVT/NRL, which occur in both sexes. In males, additional Gal-immunoreactive axons were found in many areas of the brain, with particular abundance in the dorsocaudal part of the telencephalon, thalamus, midbrain (optic tectum and tegmentum), cerebellum, and medulla oblongata (*Figure 5*). These additional axons were not observed in females, suggesting that they originate from the male-dominant sexually dimorphic *gal*-expressing neurons in pPMp.

We further evaluated the axonal projections by generating and analyzing transgenic medaka in which *gal*-expressing neurons were labeled with GFP (*gal*-GFP transgenic medaka; *Figure 5—figure supplement 1*). Fluorescence imaging of the brain of adult transgenic fish revealed intense labeling of both cell bodies in aPMp/PPa, PPp, and NAT/NVT/NRL, and axonal plexuses across the preoptic–hypothalamic region extending rostrally to the ventral telencephalon and caudally to the pituitary in both sexes. This labeling pattern was in good agreement with that determined by immunohistochemistry, suggesting that GFP expression recapitulates endogenous *gal* expression. However, GFP expression was hardly detected in the male-dominant subset of Gal-immunoreactive cell bodies and axons, and only a few cell bodies in pPMp showed weak fluorescence. As a result, the overall labeling pattern was almost identical in males and females. Although the reason for this result is not

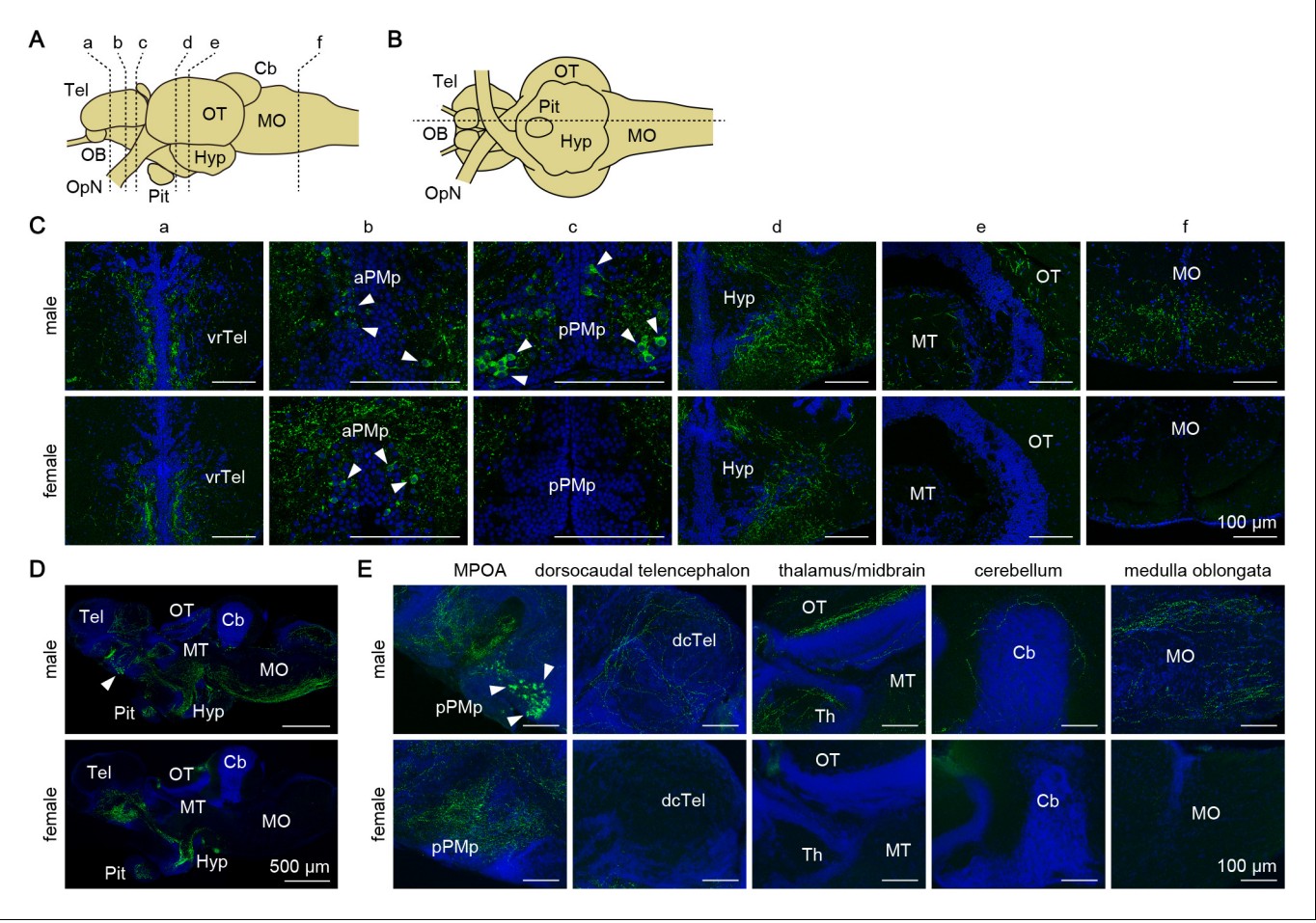

**Figure 5.** Gal peptide produced male-predominantly is transported to various brain regions. (A, B) Line drawings of lateral (A) and ventral (B) views (anterior to the left) of the medaka brain showing the approximate levels of sections in panels C and D, respectively. (C) Representative micrographs of coronal brain sections from adult males (upper panels) and females (lower panels) showing the distribution of Gal-immunoreactive cell bodies and axons. Arrowheads indicate Gal-immunoreactive neuronal cell bodies. Scale bars represent 100 μm. (D, E) Representative low (D) and high (E) magnification micrographs of sagittal brain sections (anterior to the left) from adult males (upper panels) and females (lower panels) showing the distribution of Gal-immunoreactive cell bodies and axons. Arrowheads indicate Gal-immunoreactive neuronal cell bodies in pPMp. Scale bars represent 500 μm (D) and 100 μm (E). For abbreviations of brain regions and nuclei, see *Supplementary file 1*. See also *Figure 5—figure supplement 1*. The online version of this article includes the following figure supplement(s) for figure 5:

**Figure supplement 1.** Generation and fluorescence imaging of *gal*-GFP transgenic medaka.

clear, it further supports the notion that *gal*-expressing neurons in pPMp are the primary source of the male-specific Gal-immunoreactive axons detected by immunohistochemistry.

## Gal receptors coupled to different signaling pathways are expressed widely in the brain

The sites of action of the Gal polypeptide were further delineated by identifying and characterizing genes encoding the Gal receptor (Galr) in medaka and assessing their spatial expression patterns in the brain. By screening a full-length cDNA library constructed from medaka brain, we isolated a cDNA whose deduced protein displayed high homology to GALR in other species (deposited in GenBank with accession number LC532141) (*Figure 6—figure supplement 1*). In addition, BLAST searches of public databases identified a second medaka cDNA encoding a protein highly homologous to GALR (GenBank accession numbers XM_004086657). Phylogenetic tree analysis revealed that XM_004086657 and LC532141 encoded medaka proteins orthologous to, respectively, GALR1

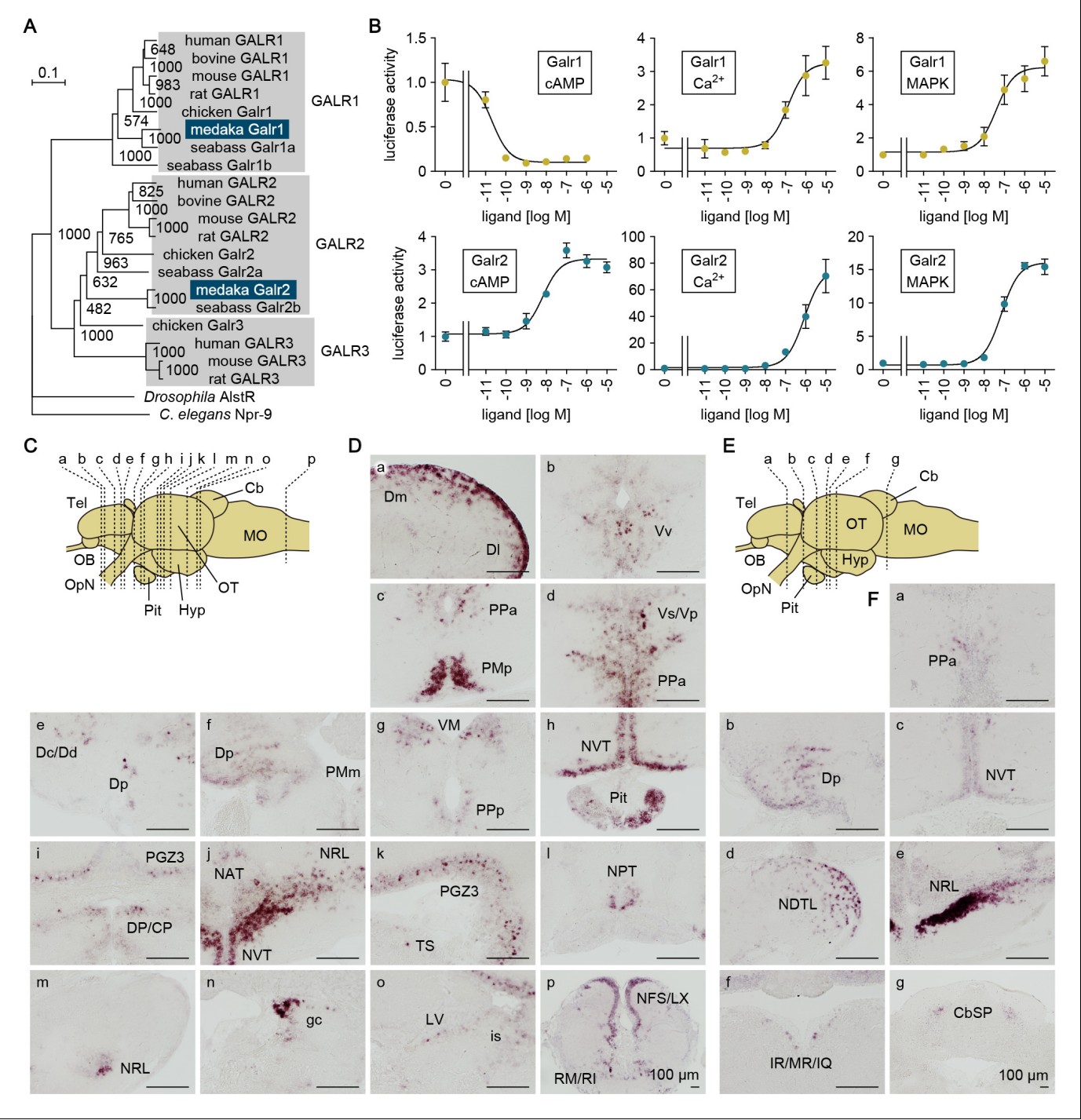

**Figure 6.** Gal receptors coupled to different signaling pathways are expressed widely in the brain. (**A**) Phylogenetic tree showing the relationship of medaka Galr1 and Galr2 to other known GAL receptors. The number at each node indicates bootstrap values for 1000 replicates. Scale bar represents 0.1 substitution per site. AlstR, allatostatin receptor. For species names and GenBank accession numbers, see *Supplementary file 2*. (**B**) Intracellular signaling pathways initiated by the activation of medaka Galr1 (upper graphs) and Galr2 (lower graphs). Cells transfected with Galr1 or Galr2 were stimulated with increasing concentrations of Gal and assayed for luciferase activity indicative of intracellular cAMP levels (left panels), $Ca^{2+}$ levels (middle panels), and MAPK (right panels) (n = 3). *x*-axis shows the concentration of Gal; *y*-axis shows the fold change in luciferase activity relative to the basal level, which was measured in the absence of Gal. (**C**) Line drawing of a lateral view (anterior to the left) of the medaka brain showing the approximate levels of sections in panel D. (**D**) Distribution of *galr1* expression in the brain and pituitary. (**E**) Line drawing of a lateral view (anterior to the left) of the medaka brain showing the approximate levels of sections in panel F. (**F**) Distribution of *galr2* expression in the brain and pituitary. All images

*Figure 6 continued on next page*

*Figure 6 continued*

are coronal sections. Only images of males are shown because there were no obvious sex differences in the distribution of expression (n = 5 per sex). Scale bars represent 100 μm. For abbreviations of brain regions and nuclei, see *Supplementary file 1*. See also *Figure 6—figure supplement 1*. The online version of this article includes the following figure supplement(s) for figure 6:

**Figure supplement 1.** Sequence information for medaka *galr2*.

and GALR2 in other species (*Figure 6A*). Tetrapods possess an additional GALR (GALR3); however, no GALR3-like gene was identified in either our cDNA library or public medaka databases, indicating the absence of GALR3 in medaka, similar to other teleosts (*Martins et al., 2014*).

Receptor activation assays showed that both Galr1 and Galr2 were activated by medaka Gal in a dose-dependent manner; however, the activation of these receptors had different consequences on downstream signaling (*Figure 6B*). Activation of Galr1 caused a 9.8-fold reduction in intracellular cAMP levels, with a Gal concentration for half-maximal effect (EC50) of 21 pM. By contrast, activation of Galr2 increased cAMP levels 3.3-fold with a much higher EC50 value of 6.8 nM. Activation of both Galr1 and Galr2 increased intracellular $Ca^{2+}$ levels with fairly similar EC50 values (130 and 800 nM, respectively), but Galr2 activation evoked a much higher maximal level as compared with Galr1 activation (76-fold versus 3.3-fold). Activation of both Galr1 and Galr2 also moderately stimulated mitogen-activated protein kinase (MAPK) activity (respectively, 6.2- and 16-fold maximal stimulation; and 40 and 72 nM EC50). In medaka, therefore, Gal signaling via Galr1 is primarily inhibitory, causing marked suppression of the cAMP pathway, whereas that via Galr2 is excitatory, mediated mainly through a robust stimulation of $Ca^{2+}$ flux. These findings are consistent with studies in mammals showing that GALR1 and GALR2 are primarily coupled to inhibitory Gi/Go and excitatory Gq/G11 proteins, respectively (*Lang et al., 2015*).

In situ hybridization revealed differential patterns of *galr1* and *galr2* expression in the medaka brain. *galr1* was expressed in the following nuclei: Dm, Dl, Dc/Dd, and Dp in the dorsal telencephalon; Vv and Vs/Vp in the ventral telencephalon; PMp, PPa, PMm, and PPp in the POA; VM and DP/CP in the thalamus; NVT, NAT, NRL, and NPT in the hypothalamus; PGZ3 in the optic tectum; TS, LV, and is in the midbrain tegmentum; gc in the brain stem; and RM/RI and NFS/LX in the medulla oblongata (*Figure 6C*). In addition, *galr1* expression was evident in the anterior pituitary (*Figure 6C*). *galr2* was expressed in the following nuclei: PPa in the POA; Dp in the dorsal telencephalon; NVT, NDTL, and NRL in the hypothalamus; IR/MR/IQ in the midbrain tegmentum; and CbSP in the cerebellum (*Figure 6D*). There were no apparent sex differences in the spatial expression patterns of either receptor.

Collectively, these results indicate that Gal can have both inhibitory and excitatory effects in the medaka brain via the activation of, respectively, Galr1 and Galr2, which are widely but differentially distributed throughout the brain.

## Discussion

In this study, we have shown that the pPMp population of medaka MPOA neurons expresses Gal nearly exclusively in males. Although not specifying the exact neuronal population, studies in several other teleosts have also demonstrated a male-biased abundance of MPOA Gal neurons (*Cornbrooks and Parsons, 1991*; *Rao et al., 1996*; *Rodríguez et al., 2003*; *Tripp and Bass, 2020*), suggesting that the sexual dimorphism in pPMp Gal neurons is conserved in teleosts. This differs substantially from the situation in mammals, where sex differences in MPOA GAL neurons are species-specific and even strain-specific in mice (*Bloch et al., 1993*; *Park et al., 1997*; *Mathieson et al., 2000*; *Wu et al., 2014*). Interestingly, the mechanism underlying sex differences in MPOA GAL neurons also seems to differ between medaka and mammals. In mammals, sex differences in MPOA GAL neurons are established during fetal life; this process is mediated by testicular testosterone, largely through its aromatase-mediated conversion in the brain to E2, which subsequently binds to ER (*Bloch et al., 1993*; *Park et al., 1997*). By contrast, we found that the sex difference in pPMp Gal neurons in medaka emerges at the onset of puberty and is mediated by testicular KT (a non-aromatizable androgen), which acts directly via AR without conversion to E2. In addition, examination of sex-reversed medaka revealed that this sex difference is independent of sex chromosome

complement and, more importantly, treatment of adult females with KT induced a significant increase in pPMp Gal neurons. This is in stark contrast to the findings in rats and ferrets, where treating adult females with testosterone/E2 did not increase GAL neurons or their axons in the MPOA (*Park et al., 1997*; *Polston and Simerly, 2003*). Therefore, the sex difference in MPOA Gal neurons in medaka depends largely or even perhaps solely on the pubertal/adult sex steroid milieu and can be reversed in adulthood, whereas that in mammals is irrevocably established early in life and, as a result, is robust in adulthood.

Furthermore, we detected the expression of *ara* in pPMp Gal neurons, suggesting that androgen directly acts on these neurons via Ara to stimulate *gal* expression. However, transgenic medaka carrying a bacterial artificial chromosome (BAC) with a GFP cassette in-frame with *gal* failed to recapitulate the male-biased expression pattern of endogenous *gal* in pPMp, although it gave an accurate representation of the sex-unbiased expression of *gal* in other brain nuclei. This result was unexpected because the BAC included a large segment of genomic DNA (54 kb upstream and 23 kb downstream of *gal*) that was considered to contain all regulatory elements for *gal* expression. It is possible that the regulatory element that directs androgen-dependent *gal* expression lies many tens or even hundreds of kilobases away from the *gal* gene body.

The question then turns to the specific role of the male-biased MPOA Gal neurons in teleosts. We demonstrated that genetic ablation of *gal* in medaka resulted in a marked reduction in aggressive chases, but no differences in other aggressive acts and parameters of mating behavior. Importantly, this behavioral deficit was observed exclusively in males, while females displayed hardly any chases or other aggressive acts regardless of their genotype. It thus seems reasonable to assume that the male-biased pPMp Gal neurons are responsible for this deficit. We further demonstrated that, while treatment of female medaka with androgen-induced male-typical aggressive behaviors, genetic ablation of *gal* in these females specifically attenuated the effect of androgen on chases. Given that pPMp Gal neurons were found to be the only population of Gal neurons that was male-biased and androgen-dependent, this result further supports the role of this neuronal population in chases. Taken together, our findings indicate that MPOA Gal neurons probably play an exclusive behavioral role in mediating male–male chases under the control of androgen/AR signaling in medaka. It seems likely that different aggressive acts are regulated by separate, albeit interacting, neural circuits and that MPOA Gal neurons may be specifically embedded within the circuit mediating chases, acting downstream of androgen/AR signaling, which seems to affect most, if not all, circuits. This assumption is compatible with the view that sexually dimorphic social behaviors are regulated in a modular manner by multiple sexually dimorphic genes acting downstream of sex steroid signaling (*Xu et al., 2012*).

Our results also show that heterozygous and homozygous males displayed similar defects in chases. Although we do not have additional data to clarify this finding, the level of the Gal peptide conferred by a single allele may be insufficient for normal male–male chases. Haploinsufficiency causing similar behavioral phenotypes in heterozygotes and homozygotes has been reported for several genes relevant to social interaction and communication, including *Shank3* (encoding an excitatory synapse scaffolding protein) and *Sh3rf2* (encoding a putative ubiquitin E3 ligase) (*Bozdagi et al., 2010*; *Wang et al., 2018*). *gal* may also be haploinsufficient in medaka with regard to its function in male–male chases. Alternatively, the mutated *gal* allele may exert a dominant-negative influence over the wild-type allele, but this seems unlikely since the mutant alleles encode functionally inert products due to introduced frameshifts in the signal peptide-coding region.

It should be noted that, in mice, MPOA GAL neurons promote the motivation to engage in parenting in part by suppressing aggression (*Wu et al., 2014*; *Kohl et al., 2018*). This is in marked contrast to our findings in medaka, where MPOA Gal neurons were found to promote aggression, suggesting that an evolutionarily conserved subset of neurons in the social behavior network can serve different, even opposite (affiliative versus agonistic), functions across species. The phenotypic variation in social behaviors across species has been assumed to be due largely to the different spatial distribution of neuropeptide expression (*O'Connell and Hofmann, 2012*). There is scant but compelling evidence that differences in neuropeptide receptor distribution can also be the cause of behavioral variation. For example, differences in the regional densities of vasopressin and oxytocin receptors, particularly in the ventral pallidum and nucleus accumbens, respectively, underlie variation in pair bonding in closely related monogamous and non-monogamous vole species (*Walum and Young, 2018*). Similarly, vasopressin and oxytocin receptor densities in the lateral septum correlate

well with variation in the propensity of different finch species to flock (*Kelly and Goodson, 2014*). Our findings demonstrate that the altered functional role of neuropeptides can underlie the behavioral variation. Of note, this functional alteration may result from differences in the sites of action of neuropeptides. We found that, in medaka, the male-biased pPMp Gal neurons project extensively to various areas of the brain, especially to the dorsocaudal part of the telencephalon, thalamus, midbrain, cerebellum, and medulla oblongata, suggesting that these areas are the primary sites of action of the Gal that is produced male-predominantly. This notion is further supported by the presence of Gal receptors in these areas. In mice, MPOA GAL neurons also project to many areas of the brain. Interestingly, however, their primary projection targets seem to be rather different from those in medaka: these neurons in mice intensely project to, for example, the anteroventral periventricular nucleus, paraventricular nucleus, dorsomedial hypothalamic nucleus, bed nucleus of the stria terminalis, and supraoptic nucleus (*Kohl et al., 2018*). This species difference suggests that the projection targets of MPOA GAL neurons have diversified during evolution to such an extent as to facilitate species variation in the functional role of GAL.

We found that, in medaka, pPMp Gal neurons project densely to Dm and Dl in the dorsocaudal part of the telencephalon, where Galr1 is most widely and abundantly expressed. Dm and Dl are considered homologous to the mammalian basolateral amygdala and hippocampus, respectively, which lack direct projections from MPOA GAL neurons (*O'Connell and Hofmann, 2011*; *Kohl et al., 2018*). There is persuasive evidence both in teleosts and in mammals that these brain regions affect the decision to engage in social behaviors, including aggression (*Demski, 2013*; *Leroy et al., 2018*; *Diaz and Lin, 2020*). Given this, Gal produced in pPMp may exert its effects on chases primarily by influencing the activity of these brain regions. In Syrian hamsters, overexpression of cAMP-responsive element-binding protein in the basolateral amygdala enhances the acquisition of conditioned defeat (*Jasnow et al., 2005*). In African cichlids (*Astatotilapia burtoni*), neural activity in Dm and Dl is negatively correlated with the frequency of aggressive territorial defense (*Weitekamp and Hofmann, 2017*; *Weitekamp et al., 2017*). Based on those observations, it is possible that Gal promotes chases by reducing the activity of these brain regions to block the acquisition of conditioned defeat via the inhibitory receptor Galr1, which robustly suppresses the cAMP pathway. The direct functional connection between pPMp Gal neurons and these brain regions may explain the difference in Gal function between medaka and mice.

What, then, is the significance of the exclusive behavioral role of Gal in male–male chases? Studies in zebrafish (*Danio rerio*) have shown that chases are unique among aggressive acts in being initiated specifically by dominant males toward their subordinates after dominance relationships are established (*Oliveira et al., 2011*; *Zabegalov et al., 2019*). In view of this, Gal may help to establish and maintain a dominance hierarchy among males by inducing chases. Here, however, we observed that male medaka perform chases immediately after the first encounter (and thus before dominance relationships are established) and mutually rather than unilaterally; therefore, this possibility seems unlikely. An alternative hypothesis is that Gal may facilitate territoriality rather than aggression per se. Chases are commonly exhibited by many different species to evict intruders from the territory. The stimulatory effect of Gal on chases without affecting other aggressive acts may be the result of an increased sense of territoriality rather than enhanced aggression to achieve dominance. This latter view is supported by the observation in African cichlids, sunfish (*Lepomis macrochirus*), and midshipman fish that males of the territorial morph exhibit higher levels of brain *gal* expression as compared with non-territorial males (*Renn et al., 2008*; *Partridge et al., 2016*; *Tripp et al., 2018*). Territoriality and associated chases are costly, requiring significant time and energy and increasing the risk of injury or death, but at the same time facilitate a monopolization of limited resources, including food and potential mates. This potential benefit may have led to strong preservation of the relevant male-biased Gal neurons in teleost evolution.

The current findings are also of interest in terms of the sexual lability of the brain in teleost fish. In teleosts, including medaka, experimental manipulations that alter the sex steroid milieu can lead to the reversal of sex-typical mating behaviors even in adulthood (*Okubo et al., 2019*). That observation has been further extended by the present finding that treating adult female medaka with androgen readily induced male-typical aggressive behaviors. Hence, brain and behavior of teleosts are highly sexually labile throughout their lifetime. Because of its behavioral role and reversible sexual dimorphism in response to sex steroids, the pPMp population of Gal neurons may be a critical

element of the long-sought neural mechanism underlying the labile nature of teleost brain and behavior.

In summary, we have demonstrated that a subset of GAL neurons within the MPOA, the most highly conserved node of the social behavior network, serve very different, even opposite, behavioral roles across species, probably due to their varying functional connections with the rest of the brain. Our findings provide a striking example of how chemically conserved neural substrates give rise to variation in social behaviors across species. Presumably, the species variation in neuronal functioning has arisen under different evolutionary selective pressures in distinct species. Because males and females of the same species also often experience different evolutionary pressures and consequently display distinct behavioral patterns, sex differences can also occur in behaviorally relevant neurons. Our findings indicate that this has indeed happened to MPOA Gal neurons in teleosts. Further comparative studies across a range of species and between sexes should help to elucidate the origin and adaptive evolution of neural mechanisms underlying social behaviors and provide novel insights into how specific social behaviors are encoded in the brain.

# Materials and methods

**Key resources table**

| Reagent type (species) or resource | Designation | Source or reference | Identifiers | Additional information |
|---|---|---|---|---|
| Gene (*Oryzias latipes*) | *gal* | this paper | GenBank:LC532140 | |
| Gene (*O. latipes*) | *galr2* | this paper | GenBank:LC532141 | |
| Gene (*O. latipes*) | *actb* | GenBank | GenBank:NM_001104808 | |
| Strain, strain background (*O. latipes*) | d-rR | NBRP Medaka | strain ID:MT837 | maintained in a closed colony over 10 years in Okubo lab |
| Genetic reagent (*O. latipes*) | *gal* knockout Δ2 line | this paper | N/A | generated and maintained in Okubo lab |
| Genetic reagent (*O. latipes*) | *gal* knockout Δ10 line | this paper | N/A | generated and maintained in Okubo lab |
| Genetic reagent (*O. latipes*) | *gal*-GFP transgenic | this paper | N/A | generated and maintained in Okubo lab |
| Cell line (*Homo sapiens*) | HEK293T | Riken BRC Cell Bank | cell number: RCB2202; RRID:CVCL_0063 | |
| Cell line (*Escherichia coli*) | DY380 | DOI:10.1038/35093556; DOI:10.1006/geno.2000.6451 | N/A | |
| Transfected construct (*H. sapiens*) | pcDNA3.1/V5-His-TOPO | Thermo Fisher Scientific | cat#:K480001 | |
| Transfected construct (*H. sapiens*) | pGL4.29 | Promega | cat#:E8471 | |
| Transfected construct (*H. sapiens*) | pGL4.33 | Promega | cat#:E1340 | |
| Transfected construct (*H. sapiens*) | pGL4.74 | Promega | cat#:E6921 | |

*Continued on next page*

*Continued*

| Reagent type (species) or resource | Designation | Source or reference | Identifiers | Additional information |
|---|---|---|---|---|
| Antibody | alkaline phosphatase-conjugated anti-DIG antibody (sheep polyclonal) | Roche Diagnostics | cat#:11093274910; RRID:AB_514497 | (1:500 or 1:2000) |
| Antibody | horseradish peroxidase-conjugated anti-fluorescein antibody (sheep polyclonal) | PerkinElmer | cat#:NEF710001EA; RRID:AB_2737388 | (1:1000) |
| Antibody | anti-GAL antibody (rabbit polyclonal) | Enzo Life Sciences | cat#:BML-GA1161-0025; RRID:AB_2051473 | (1:200 or 1:500) |
| Antibody | Alexa Fluor 555-conjugated goat anti-rabbit IgG (goat polyclonal) | Thermo Fisher Scientific | cat#:A-21428; RRID:AB_2535849 | (1:1000) |
| Antibody | Alexa Fluor 488-conjugated goat anti-rabbit IgG (goat polyclonal) | Thermo Fisher Scientific | cat#:A-11070; RRID:AB_2534114 | (1:1000) |
| Recombinant DNA reagent | full-length cDNA clone for medaka *gal* | this paper | clone ID:56_B03 | |
| Recombinant DNA reagent | full-length cDNA clone for medaka *galr2* | this paper | clone ID:39_L19 | |
| Recombinant DNA reagent | pGEM-Teasy vector | Promega | cat#:A1360 | |
| Recombinant DNA reagent | pCS2+hSpCas9 plasmid | Addgene | RRID:Addgene_51815 | |
| Recombinant DNA reagent | medaka bacterial artificial chromosome (BAC) clone containing the *gal* locus | NBRP Medaka | clone ID:108_J05 | |
| Recombinant DNA reagent | phrGFP II-1 mammalian expression vector | Agilent Technologies | cat#:240143 | |
| Sequence-based reagent | CRISPR RNA (crRNA) for medaka *gal* | Fasmac | N/A | GAGCATCGGGCTGGTTATCGCGG |
| Sequence-based reagent | trans-activating CRISPR RNA (tracrRNA) | Fasmac | cat#:GE-002 | |
| Peptide, recombinant protein | medaka Gal peptide | Scrum | this paper | GWTLNSAGYLLGPHGIDGHRTLGDKQGLA-NH$_2$ |
| Commercial assay or kit | Isogen Poly(A)$^+$ Isolation Pack | Fujifilm Wako Pure Chemical Corporation | cat#:314–05651 | |
| Commercial assay or kit | nylon membrane | Roche Diagnostics | cat#:11209272001 | |
| Commercial assay or kit | DIG RNA Labeling Mix | Roche Diagnostics | cat#:11277073910 | |
| Commercial assay or kit | T7 RNA polymerase | Roche Diagnostics | cat#:10881775001 | |

*Continued on next page*

*Continued*

| Reagent type (species) or resource | Designation | Source or reference | Identifiers | Additional information |
|---|---|---|---|---|
| Commercial assay or kit | DIG Easy Hyb | Roche Diagnostics | cat#:11603558001 | |
| Commercial assay or kit | CDP-Star | Roche Diagnostics | cat#:12041677001 | |
| Commercial assay or kit | RNeasy Lipid Tissue Mini Kit | Qiagen | cat#:74804 | |
| Commercial assay or kit | RNeasy Plus Universal Mini Kit | Qiagen | cat#:73404 | |
| Commercial assay or kit | Omniscript RT Kit | Qiagen | cat#:205111 | |
| Commercial assay or kit | SuperScript VILO cDNA Synthesis Kit | Thermo Fisher Scientific | cat#:11754050 | |
| Commercial assay or kit | Power SYBR Green PCR Master Mix | Thermo Fisher Scientific | cat#:4367659 | |
| Commercial assay or kit | LightCycler 480 SYBR Green I Master | Roche Diagnostics | cat#:04887352001 | |
| Commercial assay or kit | TSA Plus Fluorescein System | PerkinElmer | cat#:NEL741001KT | |
| Commercial assay or kit | mMessage mMachine SP6 Kit | Thermo Fisher Scientific | cat#:AM1340 | |
| Commercial assay or kit | Dual-Luciferase Reporter Assay System | Promega | cat#:E1910 | |
| Chemical compound, drug | methyltestosterone | Fujifilm Wako Pure Chemical Corporation | cat#:136–09931 | |
| Chemical compound, drug | estradiol-17β (E2) | Fujifilm Wako Pure Chemical Corporation | cat#:058–04043 | |
| Chemical compound, drug | 11-ketotestosterone (KT) | Cosmo Bio | cat#:117 ST | |
| Chemical compound, drug | tricaine methane sulfonate | Sigma-Aldrich | cat#:E10521 | |
| Chemical compound, drug | 5-bromo-4-chloro-3-indolyl phosphate | Roche Diagnostics | cat#:11383221001 | |
| Chemical compound, drug | nitro blue tetrazolium | Roche Diagnostics | cat#:11383213001 | |
| Chemical compound, drug | agarose, type IX-A | Sigma-Aldrich | cat#:A2576 | |
| Chemical compound, drug | Fast Red | Roche Diagnostics | cat#:11496549001 | |
| Chemical compound, drug | 4′,6-diamidino-2-phenylindole (DAPI) | Thermo Fisher Scientific | cat#:D1306 | |
| Chemical compound, drug | blocking reagent | Roche Diagnostics | cat#:11096176001 | |

*Continued on next page*

*Continued*

| Reagent type (species) or resource | Designation | Source or reference | Identifiers | Additional information |
|---|---|---|---|---|
| Chemical compound, drug | Lipofectamine LTX | Thermo Fisher Scientific | cat#:15338100 | |
| Software, algorithm | InterPro | https://www.ebi.ac.uk/interpro/ | RRID:SCR_006695 | |
| Software, algorithm | SignalP | http://www.cbs.dtu.dk/services/SignalP/ | RRID:SCR_015644 | |
| Software, algorithm | ClustalW | http://clustalw.ddbj.nig.ac.jp/index.php | RRID:SCR_017277 | |
| Software, algorithm | GraphPad Prism | GraphPad Software | RRID:SCR_002798 | |

## Animals

Medaka wild-type d-rR strain and its genetically modified derivatives were bred and maintained at 28°C with a 14 hr light/10 hr dark cycle. They were fed 3–4 times per day with live brine shrimp and commercial pellet food (Otohime; Marubeni Nissin Feed, Tokyo, Japan). Spawning adult fish (aged 3–4 months) were used for all analyses except for the determination of *gal* expression levels during growth and sexual maturation, where fish aged 1, 2, 3, and 7 months were used. They were randomly assigned to experimental groups. All sampling was conducted at 1–3 hr after onset of the light period.

## Production of sex-reversed medaka

Sex-reversed XX males and XY females were produced as described previously (*Okubo et al., 2011*). In brief, fertilized eggs were treated with methyltestosterone (Fujifilm Wako Pure Chemical Corporation, Osaka, Japan) at high temperature (32°C) for the production of XX males, or with E2 (Fujifilm Wako Pure Chemical Corporation) at normal temperature (28°C) for the production of XY females.

## Gonadectomy and drug treatment

For both male and female fish, the gonad was removed under tricaine methane sulfonate anesthesia (0.02%) (Sigma-Aldrich, St. Louis, MO) through a small incision made in the ventrolateral abdominal wall. Immediately after removal of the gonad, the incision was sutured with nylon thread. Sham-operated fish received the same surgical treatment as gonadectomized fish, except for removal of the gonad. After a recovery period of 3 days in saline (0.9% (w/v) NaCl), gonadectomized fish were immersed in water containing 100 ng/ml of KT (Cosmo Bio, Tokyo, Japan) or E2 (Fujifilm Wako Pure Chemical Corporation), or vehicle (ethanol) alone for 6 days and then sampled. Sham-operated fish were treated with vehicle alone and used as controls. In another experiment, females of *gal* knockout lines with intact ovaries were treated with 100 ng/ml of KT by immersion in water for 10 days and then sampled. The sex steroid concentration used was based on previously reported serum steroid levels in medaka (*Foran et al., 2002*; *Foran et al., 2004*; *Tilton et al., 2003*).

## cDNA cloning

A full-length cDNA library was constructed from the medaka brain, and approximately 32,000 clones were randomly selected and sequenced in the 5′ to 3′ direction as described previously (*Okubo et al., 2011*). After assembly and annotation, two clones each with best BLAST hits to either *gal* or *galr2* in other vertebrates were identified. One representative from each group of clones (clone ID: 56_B03 for *gal* and 39_L19 for *galr2*) was fully sequenced. To identify additional putative Galr genes in medaka, BLAST searches of the GenBank (https://www.ncbi.nlm.nih.gov/) and Ensembl (http://www.ensembl.org/index.html) databases were conducted by using known Galr sequences of other species as queries.

The deduced amino acid sequence of the putative medaka *gal* was analyzed for the presence of specific domains or motifs by using InterPro (https://www.ebi.ac.uk/interpro/) and SignalP (http://www.cbs.dtu.dk/services/SignalP/). The deduced amino acid sequences of the putative medaka *gal* and *galr1*/*galr2* genes were aligned with, respectively, GAL/galanin-like peptide (GALP) and GALR1/GALR2/GALR3 of other vertebrates by using ClustalW. The resulting alignments were used to construct bootstrapped (1000 replicates) neighbor-joining trees (http://clustalw.ddbj.nig.ac.jp/index.php). Human and mouse spexin hormones (SPX) were used as outgroups to root the GAL tree. The Allatostatin receptor (AlstR) of *Drosophila melanogaster* and galanin-like G-protein-coupled receptor Npr-9 of *Caenorhabditis elegans* were used as outgroups to root the GALR tree. The species names and GenBank accession numbers of the sequences used are listed in *Supplementary file 2*.

## Northern blot analysis

Poly(A)$^+$ RNA was isolated from the whole brains of male and female fish by using the Isogen Poly (A)$^+$ Isolation Pack (Fujifilm Wako Pure Chemical Corporation). Poly(A)$^+$ RNA (7 μg) was separated on a 1.5% denaturing agarose gel and transferred to a nylon membrane (Roche Diagnostics, Basel, Switzerland). A 529 bp fragment corresponding to nucleotides 5–533 of the medaka *gal* cDNA (GenBank accession number LC532140) was PCR-amplified, subcloned into the pGEM-Teasy vector (Promega, Madison, WI), and transcribed in vitro to generate a digoxigenin (DIG)-labeled cRNA probe for *gal* by using DIG RNA Labeling Mix and T7 RNA polymerase (Roche Diagnostics). The resulting probe was hybridized to the membrane in DIG Easy Hyb (Roche Diagnostics) at 65°C overnight. Hybridization signals were detected with alkaline phosphatase-conjugated anti-DIG antibody and CDP-Star (Roche Diagnostics). Image acquisition was performed by using a LAS-3000 Mini chemiluminescence imaging system (Fujifilm, Tokyo, Japan).

## Real-time PCR

Total RNA was isolated from whole brain by using the RNeasy Lipid Tissue Mini Kit or RNeasy Plus Universal Mini Kit (Qiagen, Hilgen, Germany). Complementary DNA was synthesized by using the Omniscript RT Kit (Qiagen) or SuperScript VILO cDNA Synthesis Kit (Thermo Fisher Scientific, Waltham, MA). Real-time PCR to determine levels of *gal* expression was performed either on the ABI Prism 7000 Sequence Detection System using the Power SYBR Green PCR Master Mix (Thermo Fisher Scientific) or on the LightCycler 480 System II using the LightCycler 480 SYBR Green I Master (Roche Diagnostics). For every reaction, melting curve analysis was conducted to ensure that a single amplicon was produced in each sample. Levels of β-actin (*actb*) expression in each sample were used for normalization. The primers used for real-time PCR are listed in *Supplementary file 3*.

## Single-label in situ hybridization

The 1232- and 1276 bp fragments corresponding to nucleotides 491–1722 of the medaka *galr1* cDNA (GenBank accession number XM_004086657) and 7–1282 of the medaka *galr2* cDNA (LC532141) were PCR-amplified and used to generate DIG-labeled probes for *galr1* and *galr2*, respectively, as described above.

The procedure for single-label in situ hybridization has been described previously (*Hiraki et al., 2012*). In brief, whole brains were fixed in 4% paraformaldehyde (PFA) and embedded in paraffin. Serial coronal sections of 10 μm thickness were cut and hybridized with the DIG-labeled *galr1* or *galr2* probe or the same DIG-labeled *gal* probe used for Northern blot analysis. Hybridization signals were visualized by using alkaline phosphatase-conjugated anti-DIG antibody and 5-bromo-4-chloro-3-indolyl phosphate/nitro blue tetrazolium (BCIP/NBT) substrate (Roche Diagnostics). Color development was allowed to proceed for more than 5 hr, or was stopped after 15 min (for quantification of *gal* expression in males that were castrated and treated with sex steroids) or 1 hr (for quantification of *gal* expression in females that were ovariectomized and treated with sex steroids) to avoid saturation. All sections in each comparison were processed simultaneously under the same conditions. For *galr1*, for which hybridization signals were detected throughout the brain and thus the specificity of the staining was somewhat uncertain, in situ hybridization was also performed using a sense probe under conditions identical to those used for the antisense probe. No signal was obtained by the sense probe, confirming that the signals obtained by the antisense probe represented specific staining.

To obtain quantitative data, the number of *gal*-expressing neurons in each brain nucleus was counted manually. In some brain regions, *gal* neurons were continuously distributed across two or more nuclei such that it was difficult to count the number of the neurons separately in each nucleus. In this case, the total number of the neurons in the nuclei was counted. Only intensively labeled neurons with a clearly defined cell body were counted to reduce, if not eliminate, the likelihood of double counting of the neurons split between sections. Although the resulting counts may still slightly overestimate the actual number of the neurons, the same counting procedure was used throughout, so that the results are comparable. Medaka display a sex difference in mean adult body size, with females being larger than males. The mean weight of the adult female brains was also found to be significantly greater than that of the adult male brains (data not shown). However, since the difference was only 1.12-fold and much smaller than the individual variation, the sex difference in brain size was not controlled for in examining the number of the neurons. Brain nuclei were identified by using medaka brain atlases (*Anken and Bourrat, 1998*; *Ishikawa et al., 1999*), supplemented with information from Nissl-stained sections (*Kawabata et al., 2012*).

## Double-label in situ hybridization

Double-label in situ hybridization was performed as described previously (*Takeuchi and Okubo, 2013*). In brief, whole brains were fixed in 4% PFA and embedded in 5% agarose (Type IX-A; Sigma-Aldrich) supplemented with 20% sucrose. Frozen 20 µm thick coronal sections were cut and hybridized simultaneously with the DIG-labeled *gal* probe described above and a fluorescein-labeled AR (*ara*, NM_001104681; *arb*, NM_001122911) or ER (*esr1*, XM_020714493; *esr2a*, NM_001104702; *esr2b*, NM_001128512) probe described previously (*Hiraki et al., 2012*). The DIG-labeled probe was visualized by using alkaline phosphatase-conjugated anti-DIG antibody and Fast Red (Roche Diagnostics); the fluorescein-labeled probe was visualized by using horseradish peroxidase-conjugated anti-fluorescein antibody and the TSA Plus Fluorescein System (PerkinElmer, Waltham, MA). Cell nuclei were counterstained with 4′,6-diamidino-2-phenylindole (DAPI). Fluorescent images were acquired by using a confocal laser scanning microscope (Leica TCS SP8; Leica Microsystems, Wetzlar, Germany). The following excitation and emission wavelengths were used for detection: DAPI, 405 nm and 410–480 nm, respectively; fluorescein, 488 nm and 495–545 nm, respectively; and Fast Red, 552 nm and 620–700 nm, respectively.

Generation of knockout medaka *gal* knockout medaka were generated by CRISPR/Cas9-mediated genome editing. A CRISPR RNA (crRNA) was designed to target the second exon, which encodes the signal peptide of the Gal precursor protein (*Figure 3—figure supplement 1*). The crRNA and trans-activating CRISPR RNA (tracrRNA) were chemically synthesized by Fasmac (Kanagawa, Japan). Cas9 mRNA was synthesized by in vitro transcription of the linearized pCS2+hSpCas9 plasmid (Addgene plasmid number 51815; Addgene, Cambridge, MA) using the mMessage mMachine SP6 Kit (Thermo Fisher Scientific). The crRNA, tracrRNA, and Cas9 mRNA were co-microinjected into the cytoplasm of embryos at the one-cell stage. Potential founders were screened by outcrossing to wild-type fish and testing progeny for mutations at the target site using a mismatch-sensitive T7 endonuclease I assay (*Kim et al., 2009*) followed by direct sequencing. Two founders were identified that yielded a high proportion of progeny carrying deletions that caused frameshifts leading to premature truncation of the Gal precursor protein: the progeny of one founder carried a 2 bp deletion (Δ2) and progeny of the other carried a 10 bp deletion (Δ10). These progeny were intercrossed to establish knockout lines (Δ2 and Δ10 lines). Each line was maintained by breeding heterozygous males and females to obtain wild-type, heterozygous, and homozygous siblings for use in behavioral tests. The genotype of each fish was determined by direct sequencing using the primers listed in *Supplementary file 3*.

## Mating behavior test

The mating behavior of medaka consists of a sequence of stereotyped actions that are easily quantified. The sequence begins with the male approaching and following the female closely. The male then performs a courtship display, in which he swims quickly in a circular pattern in front of the female. If she is receptive, the male grasps her with his fins (termed 'wrapping'), and they quiver together ('quivering') until eggs and sperm are released ('spawning'). If the female is not receptive,

she either rapidly moves away from the male or assumes a rejection posture (*Walter and Hamilton, 1970*).

The mating behavior test was performed essentially as described elsewhere (*Hiraki-Kajiyama et al., 2019*). In brief, on the day before behavioral testing, each focal male and female (from ∆2 and ∆10 knockout lines) was placed with a stimulus fish of the opposite sex (wild-type d-rR strain) in a 2-l rectangular tank, separated by a perforated transparent partition. The partition was removed 1 hr after onset of the light period of the following day, and fish were allowed to interact for 10 min. All interactions were recorded with a digital video camera (iVIS HF S11, Canon, Tokyo, Japan, or HC-W870M, Panasonic, Osaka, Japan).

In each test, wild-type, heterozygous, and homozygous siblings reared under the same conditions were used as the comparison group to control for genetic and environmental factors. The percentage of females that spawned within the test period was calculated for each genotype. The following behavioral parameters were also calculated from the video recordings: latency to the first following by the male, the first wrapping, and the wrapping that resulted in spawning; number of courtship displays prior to spawning and wrapping attempts refused by the female; and duration of quivering. Females that did not spawn were not included in the analysis of these parameters.

## Aggressive behavior test

The aggressive behavior of teleosts, including medaka, involves the following five types of behavioral act that can be reliably identified: (1) 'chase', whereby the fish swims rapidly toward the opponent, causing the latter to flee; (2) 'fin display', whereby two fish present their lateral sides toward each other while spreading their fins; (3) 'circle', whereby two fish orient head-to-tail and rapidly circle each other with their fins spread; (4) 'strike', whereby the fish strikes the opponent with its caudal fin; and (5) 'bite', whereby the fish snaps at the opponent with its teeth (*Oliveira et al., 2011*; *Kagawa, 2013*).

All behavioral procedures were conducted in 2-l rectangular tanks contained within a large recirculating water system with a constant influx of dechlorinated tap water. Fish of each sex and genotype were randomly distributed into four tanks a week before behavioral testing. Four fish of the same sex and genotype (which were all taken from different tanks and thus did not encounter one another for a week), with each separated from the others by opaque partitions, and allowed to acclimatize to the tank for 10 min. The partitions were then removed, and the fish were allowed to interact for 20 min while their behavior was recorded as described above. All recordings were taken 1–2.5 hr after onset of the light period.

In each test, wild-type, heterozygous, and homozygous siblings reared under the same conditions were used as the comparison group. The total number of each aggressive act (chase, fin display, circle, strike, and bite) performed by the four fish in the tank was counted manually from the video recordings. In addition, the total number of chases by the four fish in the tank was calculated for each 5 min interval during the test period. The number of chases initiated and received by each individual fish was also calculated. Analysis of the video recordings at reduced speeds allowed us to track individual fish and quantify their behaviors.

## Double labeling with immunohistochemistry and in situ hybridization

A rabbit anti-GAL antibody that recognizes teleost Gal with high specificity (*Rodríguez et al., 2003*; *Amano et al., 2009*) was obtained from Enzo Life Sciences (Farmingdale, NY; RRID:AB_2051473). The specificity of this antibody was confirmed in medaka brain by a double-labeling experiment using immunohistochemistry and in situ hybridization, essentially as described elsewhere (*Hiraki-Kajiyama et al., 2019*) with slight modifications. In brief, frozen coronal brain sections of 20 µm thickness were cut as described above and hybridized with the above-mentioned *gal* probe, which was labeled with fluorescein by using Fluorescein RNA Labeling Mix and T7 RNA polymerase (Roche Diagnostics). After blocking with phosphate-buffered saline (PBS) containing 2% normal goat serum, the sections were incubated overnight at 4°C with anti-GAL antibody diluted 1:200 in PBS containing 2% normal goat serum, 0.1% bovine serum albumin, and 0.02% keyhole limpet hemocyanin. The sections were then reacted overnight at 4°C with horseradish peroxidase-conjugated anti-fluorescein antibody (PerkinElmer) and Alexa Fluor 555-conjugated goat anti-rabbit IgG (Thermo Fisher Scientific), both diluted 1:1000 in Tris-buffered saline (TBS) containing 1.5% blocking reagent (Roche

Diagnostics) and DAPI. The anti-fluorescein antibody was visualized by using the TSA Plus Fluorescein System (PerkinElmer). Fluorescent images were obtained as described above. The excitation and emission wavelengths for Alexa Fluor 555 were 552 nm and 620–700 nm, respectively.

### Immunohistochemistry

Whole brains, with pituitaries attached, were fixed in Bouin's solution and embedded in 5% agarose (Type IX-A; Sigma-Aldrich) supplemented with 20% sucrose. Frozen 40-μm-thick sections were cut in the coronal or sagittal plane. After blocking with PBS containing 2% normal goat serum, the sections were incubated overnight at 4°C with anti-GAL antibody diluted at 1:500 in PBS containing 2% normal goat serum, 0.1% bovine serum albumin, and 0.02% keyhole limpet hemocyanin. The sections were then reacted overnight at 4°C with Alexa Fluor 488-conjugated goat anti-rabbit IgG (Thermo Fisher Scientific) diluted 1:1000 in PBS. Fluorescent images were obtained as described above. Z-stack confocal images were acquired at 1 μm intervals throughout the depth of the specimen to help visualize the trajectory of the labeled axons.

### Generation of transgenic medaka

A medaka BAC clone (clone ID: 108_J05) containing the *gal* locus was obtained from the National BioResource Project (NBRP) Medaka and modified by homologous recombination in *Escherichia coli* strain DY380, essentially as described previously (*Copeland et al., 2001*; *Lee et al., 2001*). A 7 bp sequence containing the translation initiation site of *gal* in this BAC clone was replaced by a 2136 bp DNA cassette containing the humanized *Renilla reniformis* GFP II-coding sequence (Agilent Technologies, Santa Clara, CA), bovine growth hormone polyadenylation signal, and kanamycin resistance gene (*Figure 5—figure supplement 1*). The resulting BAC transgene was microinjected into the cytoplasm of embryos at the one-cell stage. Transgenic founders were screened by outcrossing to wild-type fish and examining progeny embryos for GFP fluorescence. One founder was identified that produced progeny expressing GFP in a pattern similar to endogenous *gal* expression during embryonic development. These progeny were raised to adulthood and intercrossed to establish homozygous transgenic lines.

### GFP imaging

Whole brains, with pituitaries attached, were removed from the *gal*-GFP transgenic medaka, fixed in 4% PFA, and embedded in 5% agarose (Type IX-A; Sigma-Aldrich) supplemented with 20% sucrose. Frozen 40 μm thick sections were cut in the coronal or sagittal plane. Cell nuclei were counterstained with DAPI. Fluorescent images were obtained as described above. The excitation and emission wavelengths for GFP were 488 nm and 495–545 nm, respectively.

### Receptor activation assay

The medaka Gal polypeptide with amidated C termini was synthesized by Scrum (Tokyo, Japan). The cDNA fragments encoding the full-length Galr1 and Galr2 were PCR-amplified and subcloned into the expression vector pcDNA3.1/V5-His-TOPO (Thermo Fisher Scientific). Either of the resulting Galr1 or Galr2 expression construct was transiently transfected into HEK293T cells together with a luciferase reporter vector containing *cis*-acting elements responsive to cAMP (pGL4.29; Promega), $Ca^{2+}$-dependent nuclear factor of activated T-cells (NFAT) (pGL4.30; Promega), or the MAPK signaling pathway (pGL4.33; Promega) and the internal control vector pGL4.74 (Promega) by using Lipofectamine LTX (Thermo Fisher Scientific). Forty-two hours after transfection, cells were stimulated with Gal polypeptide at doses of 0, $10^{-11}$, $10^{-10}$, $10^{-9}$, $10^{-8}$, $10^{-7}$, $10^{-6}$, and $10^{-5}$ M for 6 hr. Assay of cAMP-mediated luciferase activity was performed in the presence of 5 μM forskolin. After cell lysis, luciferase activity was measured by using the Dual-Luciferase Reporter Assay System (Promega). Each assay was performed in triplicate and repeated three times independently. HEK293T cells used in this study were authenticated by short tandem repeat profiling (National Institute of Biomedical Innovation, Osaka, Japan) and confirmed to be mycoplasma free (Biotherapy Institute of Japan, Tokyo, Japan).

## Statistical analysis

In cases where there were fewer than seven data points per experimental group, individual data points were plotted with the mean ± standard error of the mean (SEM); in cases where there were seven or more data points per experimental group, data were plotted as box and whisker plots by the Tukey method for clarity. In real-time PCR analysis, the expression level of *gal* (normalized to that of *actb*) in female whole brain was arbitrarily set to 1, and the relative difference was calculated in each data set to facilitate comparisons among data sets.

Statistical analyses were performed by using GraphPad Prism (GraphPad Software, San Diego, CA). Differences in continuous variables between two groups were analyzed by using unpaired two-tailed Student's *t*-test. Welch's correction was applied if the F-test indicated that the variance differed significantly between groups. Differences in continuous variables between more than two groups were assessed by using one-way analysis of variance (ANOVA) followed by Bonferroni's post hoc test. Homogeneity of variance was verified for all data sets by using Brown-Forsythe test. Two-way ANOVA followed by Bonferroni's post hoc test was used for analyses of *gal* expression during growth/sexual maturation, the number of chases for each 5 min interval, and the number of aggressive acts of KT-treated females. Differences in categorical variables were assessed by using Fisher's exact test.

## Acknowledgements

We thank NBRP Medaka for providing the BAC clone used in this study and Drs. Lino Tessarollo and Donald L Court for the DY380 bacteria. We also thank Dr. Shin-ichi Higashijima for the technical advice on generating the transgenic construct; Dr. Guro K Sandvik for help with the transgenic fish; and Akira Hirata for assistance with medaka husbandry.

## Additional information

### Funding

| Funder | Grant reference number | Author |
| --- | --- | --- |
| Japan Society for the Promotion of Science | 17J08702 | Junpei Yamashita |
| Japan Society for the Promotion of Science | 16H04979 | Kataaki Okubo |
| Ministry of Education, Culture, Sports, Science, and Technology | 17H06429 | Kataaki Okubo |
| Japan Society for the Promotion of Science | 19H03044 | Kataaki Okubo |

The funders had no role in study design, data collection and interpretation, or the decision to submit the work for publication.

### Author contributions

Junpei Yamashita, Conceptualization, Formal analysis, Funding acquisition, Validation, Investigation, Visualization, Methodology, Writing - original draft, Writing - review and editing; Akio Takeuchi, Kohei Hosono, Thomas Fleming, Investigation, Methodology, Writing - review and editing; Yoshitaka Nagahama, Conceptualization, Supervision, Writing - review and editing; Kataaki Okubo, Conceptualization, Formal analysis, Supervision, Funding acquisition, Validation, Investigation, Visualization, Methodology, Writing - original draft, Writing - review and editing

### Author ORCIDs

Kataaki Okubo https://orcid.org/0000-0002-4178-3094

## Ethics

Animal experimentation: All animal procedures were performed in accordance with the guidelines of the Institutional Animal Care and Use Committee of the University of Tokyo. The committee requests the submission of an animal-use protocol only for use of mammals, birds, and reptiles, in accordance with the Fundamental Guidelines for Proper Conduct of Animal Experiment and Related Activities in Academic Research Institutions under the jurisdiction of the Ministry of Education, Culture, Sports, Science and Technology of Japan (Ministry of Education, Culture, Sports, Science and Technology, Notice No. 71; June 1, 2006). Accordingly, we did not submit an animal-use protocol for this study, which used only teleost fish and thus did not require approval by the committee.

## Decision letter and Author response

Decision letter https://doi.org/10.7554/eLife.59470.sa1
Author response https://doi.org/10.7554/eLife.59470.sa2

## Additional files

### Supplementary files

• Supplementary file 1. Abbreviations of medaka brain regions and nuclei.

• Supplementary file 2. Species names and GenBank accession numbers of the protein sequences used in this study.

• Supplementary file 3. Primers used in this study.

• Transparent reporting form

### Data availability

Sequence data have been deposited in DDBJ/EMBL/GenBank with accession numbers LC532140 and LC532141.

The following datasets were generated:

| Author(s) | Year | Dataset title | Dataset URL | Database and Identifier |
|---|---|---|---|---|
| Okubo K | 2020 | Oryzias latipes gal mRNA for galanin precursor | https://www.ncbi.nlm.nih.gov/nuccore/LC532140 | NCBI GenBank, LC532140 |
| Okubo K | 2020 | Oryzias latipes galr2 mRNA for galanin receptor 2 | https://www.ncbi.nlm.nih.gov/nuccore/LC532141 | NCBI GenBank, LC532141 |

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
