## [Decision Letter]

**Acceptance summary:**

This manuscript provides an important contribution to the field of behavioral neuroscience. It elegantly blends genetics, hormonal manipulations, and genome editing to uncover the role of the neuropeptide galanin in social behavior. This work shows that galanin is regulated by androgens and governs specific aspects of aggression in medaka fish, a fascinating juxtaposition with the established role of galanin in parental behavior in rodents. This work highlights that the role of galanin in mediating social behavior is evolutionarily flexible and varies with life history strategy across species, opening the door for many other studies on the neuroendocrine regulation of behavior and its evolutionary trajectories.

**Decision letter after peer review:**

Thank you for submitting your article "Male-predominant galanin mediates androgen-dependent aggressive chases in medaka" for consideration by *eLife*. Your article has been reviewed by three peer reviewers, including Laruen A O’Connell as the Reviewing Editor and Reviewer #1, and the evaluation has been overseen by Catherine Dulac as the Senior Editor. The following individual involved in review of your submission has agreed to reveal their identity: John Godwin (Reviewer #2).

The reviewers have discussed the reviews with one another and the Reviewing Editor has drafted this decision to help you prepare a revised submission.

Summary:

This manuscript provides an important contribution to the field of behavioral neuroscience. It elegantly blends genetics, hormonal manipulations, and genome editing to uncover the role of the neuropeptide galanin in social behavior. This work shows that galanin is regulated by androgens and governs specific aspects of aggression in medaka fish, a fascinating juxtaposition with the established role of galanin in parental behavior in rodents. This work highlights that the role of galanin in mediating social behavior is evolutionarily flexible and varies with life history strategy across species, opening the door for many other studies on the neuroendocrine regulation of behavior and its evolutionary trajectories.

Essential revisions:

1) Statistical tests used should be reported in the Results section in addition to figure legends, including degrees of freedom and the test statistic, not just the p-value.

2) Please include more details regarding the cell quantification method used. How did the authors avoid confounding factors, such as the danger of counting the same cell across multiple sections? Consider also accounting for significant differences in male and female body size (if present).

---

## [Author Response]

Essential revisions:1) Statistical tests used should be reported in the Results section in addition to figure legends, including degrees of freedom and the test statistic, not just the p-value.

In response to this comment, we have added detailed information regarding the statistical tests used, including degrees of freedom and the test statistic, to the Results section.

2) Please include more details regarding the cell quantification method used. How did the authors avoid confounding factors, such as the danger of counting the same cell across multiple sections? Consider also accounting for significant differences in male and female body size (if present).

Although we counted only intensively labeled neurons with a clearly defined cell body, we cannot exclude the possibility that the same neuron was counted twice in adjacent sections. Therefore, our neuronal counts may overestimate the actual neuronal number due to occasional double counting. However, since the same counting procedure was used throughout, we believe that the results are comparable.

Medaka display a sex difference in mean adult body size, with females being larger than males. We found that the mean weight of the adult female brains was significantly higher than that of the adult male brains (please see Author response image 1). However, since the difference was only 1.12-fold and much smaller than the individual variation in either sex, we did not control for sex differences in brain size.

These pieces of information have been included in the Materials and methods section as follows: “Only intensively labeled neurons with a clearly defined cell body were counted to reduce, if not eliminate, the likelihood of double counting of the neurons split between sections. […] However, since the difference was only 1.12-fold and much smaller than the individual variation, the sex difference in brain size was not controlled for in examining the number of the neurons.”.